# Population Dynamics of *Bactrocera dorsalis* (Diptera: Tephritidae) in Four Counties of Yunnan, China, by Electronic Monitoring System

**DOI:** 10.3390/insects15080621

**Published:** 2024-08-18

**Authors:** Ziyuan Li, Yan Li, Yuling Liang, Yixiang Qi, Yongyue Lu, Jiao Ma

**Affiliations:** 1College of Plant Protection, South China Agricultural University, Guangzhou 510642, China; ziyuanli@stu.scau.edu.cn (Z.L.); liangyuling@stu.scau.edu.cn (Y.L.); qiyixiang19880922@163.com (Y.Q.); 2Yunnan Plant Protection and Quarantine Station, Kunming 650034, China; liyan7603@163.com; 3Honghe Academy of Agricultural Sciences, Honghe 651400, China

**Keywords:** population dynamics, Yunnan, automated monitoring system, intelligent plant protection

## Abstract

**Simple Summary:**

*Bactrocera dorsalis* (Hendel) is a serious threat to the fruit industry. Implementing monitoring procedures is essential for effective pest control. The monitoring of *B. dorsalis* is predominantly conducted through the use of yellow sticky traps and baited traps, followed by manual counting. However, this approach is both time consuming and labor intensive. In this study, we employed automated identification and counting devices for monitoring the adult population dynamics of *B. dorsalis* in four counties with disparate climatic types in Yunnan. The differing climatic types and temperature fluctuations resulted in disparate population dynamics of *B. dorsalis* in the four counties, yet there were two peaks of adult emergence in each year. Furthermore, the population of *B*. *dorsalis* was correlated with temperature. The automated monitoring system employed in the study permitted the daily monitoring of adult *B. dorsalis* populations, thereby facilitating the tracking of the daily dynamics of *B. dorsalis* by farmers and researchers alike.

**Abstract:**

*Bactrocera dorsalis* (Hendel) (Diptera: Tephritidae) is a global economic pest that poses a serious threat to the fruit industry. In the southwest of China, Yunnan Province sustains a severe infestation of *B. dorsalis*. An automated monitoring system designed for *B. dorsalis* was employed in this study to elucidate the annual population dynamics of *B. dorsalis* in four counties: Yuanjiang, Huaping, Guangnan, and Ludian in Yunnan. The system utilizes sex parapheromone and image recognition technology. The data uploaded by the device are used to analyze the annual population dynamics of *B. dorsalis* in different regions. The results showed that the populations of adult *B. dorsalis* in all four counties peaked twice annually, with Yuanjiang experiencing the earliest peak periods, followed by Huaping, Guangnan, and Ludian. Adult *B. dorsalis* occurred in Yuanjiang throughout the year, and Yuanjiang had the highest number of *B. dorsalis* monitored. In Huaping, adult *B. dorsalis* occurred in March–December and was highly active, with a high population density in 2019. *Bactrocera dorsalis* did not occur in December in Guangnan but only in May–October in Ludian. *Bactrocera dorsalis* abundance was correlated with temperature in all four areas. The outcomes of this experiment provide a practical foundation for developing control strategies targeting *B. dorsalis* in various orchards across each county.

## 1. Introduction

*Bactrocera dorsalis* is a global pest responsible for millions of annual crop losses and is expensive to manage [1,2,3]. This species affects a variety of fruits, including citrus [4], mango [5], and apple [1]. Due to its polyphagous feeding habits [6], *B. dorsalis* has rapidly colonized new regions, rendering it challenging to control [3,7]. Additionally, *B. dorsalis* can survive in temperatures ranging from 15 °C to 33 °C [8], and due to global trade and global warming, its suitable habitat has expanded worldwide [9]. After it was first reported in Taiwan, *B. dorsalis* gradually appeared on some islands in the Asia-Pacific region, then quickly invaded Africa, and it was also found in Oceania and the Americas. The area affected by *B. dorsalis* accounts for 52.4% of the total area of Asia, making this continent the most severely affected worldwide [10].

In China, *B. dorsalis* has invaded temperate regions, such as Beijing, where the adult population density is the highest from August to October, with 2–3 generations of *B. dorsalis* per year [11]. For instance, there are 3–5 generations of *B. dorsalis* per year in subtropical regions like Shanghai, while *B. dorsalis* completes 10 generations annually in Hainan Province and Guangdong Province in tropical areas of China [9].

Reliable monitoring of pest populations is crucial for understanding the relationship between pest behavior and ecology in invaded regions. Monitoring provides detailed information on the densities of pest population fluctuations and changes in their distribution over time, which is essential for formulating effective pest management strategies [12]. Currently, the majority of population dynamic monitoring of *B. dorsalis* or other Tephritid fruit flies utilizes either sex parapheromone traps or yellow sticky traps with manual counting [7,13,14,15]. However, this method is time consuming and labor intensive, requiring manual counting.

Several studies have been conducted on the automatic identification and monitoring of Tephritid pests. For instance, researchers enhanced plastic traps with optoelectronic sensors that analyze the wingbeat frequency and pattern of the Tephritid fruit flies [16]. Similarly, another monitoring system employs the optoelectronic sensors that detect the signals caused by the partial occlusion of infrared light due to the wingbeats of the fruit flies [17]. An automated monitoring system for *B. dorsalis*, incorporating meteorological monitoring and utilizing methyl eugenol, a male attractant for flies of the genus *Bactrocera* [18], combined with infrared counting technology, has been deployed across the entire Taiwan Province [19]. Digital image processing and recognition play a crucial role in smart agriculture and intelligent plant protection through the development of artificial intelligence. This technology enables real-time monitoring of crop diseases, pests, and weeds, guiding the implementation of control measures [20,21,22,23,24]. An electronic trap that utilized real-time image transmission technology in the Mediterranean was able to monitor different spices of important Tephritid fruit fly pests such as *Ceratitis capitata*, *Dacus ciliatus*, *Rhagoletis cerasi,* and *B*. *oleae* [25]. Image recognition technology is advancing the automation of pest monitoring, enabling high discrimination.

Yunnan Province, which is situated at the southern end of mainland China and shares a border with Myanmar, Vietnam, and Laos, coupled with the varied crops and fruits grown, creates an environment conducive to the introduction and proliferation of invasive alien species [26,27,28,29]. Yunnan experiences the most severe occurrence of *B. dorsalis*, leading to an annual 30% reduction in mango yield in Yuanjiang County, the highest mango-producing county in Yunnan [30]. To elucidate the temporal and spatial patterns of adult *B. dorsalis* infestation across diverse climatic regimes and orchard typologies within Yunnan Province, an electronic monitoring system was used to record and track the daily population dynamics of *B. dorsalis* in orchards in four counties in Yunnan, Yuanjiang, Guangnan, Huaping, and Ludian from 2018 to 2019. In order to further understand the impact of meteorological factors on the occurrence of *B. dorsalis*, the daily dynamics of the pest were analyzed in conjunction with meteorological data from the same period. These findings provide a scientific foundation for the development of control strategies for *B. dorsalis* in different regions of Yunnan Province.

## 2. Materials and Methods

### 2.1. Study Area

The population dynamics of *B. dorsalis* were investigated in Yunnan, China, from 2018 to 2019. Daily monitoring of *B. dorsalis* was conducted in mango orchards in Yuanjiang and Huaping, a citrus orchard in Guangnan, and an apple orchard in Ludian, representing diverse climatic conditions and altitudes (Figure 1). Yuanjiang County is characterized by a dry-hot valley climate, with an annual average temperature of 23.8 °C. Huaping County experiences a subtropical low-heat valley with an annual average temperature of 19.8 °C. Due to the fact that both Yuanjiang and Huaping are situated within valley climates, they exhibit relatively low annual average precipitation, with figures of 543.9 mm and 403 mm, respectively. Ludian has a low-latitude plateau monsoon climate, while Guangnan has a low-latitude plateau subtropical monsoon climate. Their annual average temperatures are 12.1 °C and 18 °C, and their annual precipitation is 893.9 mm and 765 mm, which is higher than that of Huaping and Yuanjiang.

### 2.2. Monitoring the Population Dynamics of B. dorsalis

This study used an electronic monitoring system (TYPE RTJK-1S) manufactured by Guangzhou Ruifeng Biotechnology Co., Ltd. (Guangzhou, China) (Figure 2). The system relies on sex parapheromone technology, combined with photoelectric induction and Internet of Things technology, to develop a remote real-time pest reporting system (http://www.ruifengbio.com/products/zhihuinongye/1258.html (accessed on 6 November 2023)).

The monitoring system, powered by solar energy with a system voltage of 12 V. Methyl eugenol (ME, produced by Guangzhou Ruifeng Biotechnology Co., Ltd. (Guangzhou, China)) was replenished monthly in the device since it is a volatile compound. The monitoring system uses a TTL serial camera with a resolution range of 200–1200 W. The imaging process is based on an STM32 microcontroller software (RF-XY-V2.0) control system. The recognition process is based on the Visual Transformer, which serves as the backbone network for the recognition method. This configuration guarantees the integrity and accuracy of data acquisition within the monitoring system. The Internet of Things technology empowers the system to interconnect sensors and devices implanted in an orchard environment to the internet, facilitating the real-time collection and transmission of data to distant servers or monitoring hubs. This configuration permits researchers or operators to retrieve the data from any location through the network, facilitating real-time surveillance of pest activities within the orchard.

Although the electronic monitoring system is already a commercially available product, its reliability was assessed in Yuanjiang County from 21 March to 18 April 2017. An electronic monitoring system and three bottle traps (Figure 3) were set up in a mango orchard at an altitude of 420 m and a peach orchard at an altitude of 1600 m. Daily counts of trapped flies were conducted. The monitoring system and the bottle traps were placed 50 m apart, arranged in a square configuration. This setup allowed for a comparative analysis of the electronic system’s effectiveness against traditional trapping methods.

During 2018–2019, the monitoring system was set up to count *B. dorsalis* once a day in this study. A monitoring device was stationed at each monitoring site within every county. The monitoring devices placed at a distance of 1 m from the fruit trees, with the positioning ensuring that the devices remained unobstructed by the tree canopies.

### 2.3. Data Analysis

The population dynamics of *B. dorsalis*, as observed through manual and electronic monitoring systems, were analyzed using Origin 2021 software [34]. Additionally, a linear correlation analysis was conducted to compare the results from both monitoring methods. The “Simple Fit” tool in Origin 2021 was used for this analysis and for visualizing the outcomes.

We downloaded the data for each monitoring site from the online digital plant protection big data platform of Guangzhou Ruifeng Biotechnology Co., Ltd. (Guangzhou, China). Meteorological data, including daily low and high temperatures, precipitation, wind speed, and humidity, were downloaded from the China Meteorological Data Network (https://data.cma.cn/ (accessed on 23 July 2024)). The number of individuals collected per day was plotted along with the mean daily temperature and daily total precipitation using Origin 2021. We also visualized temperature and insect counts to determine the temperature range where *B. dorsalis* activity peaks.

To investigate the relationship between daily *B. dorsalis* counts and environmental factors such as temperature, precipitation, humidity, and wind speed, Pearson’s correlation analysis was performed. The correlation matrix was plotted using the R package “corrplot” [35], and a Bonferroni correction was applied to account for multiple comparisons. We also conducted a principal component analysis (PCA) using the “prcomp” function. Correlation coefficients were calculated to support our PCA findings and visualized using the “corrplot” package. The outcomes of the PCA were graphically illustrated using the “fviz_pca_biplot” function from the “factoextra” package [36].

## 3. Results

### 3.1. Accuracy Assessment of Automated Monitoring System

In the surveillance conducted from March to April 2017 across two orchards, the trends in the adult population dynamics of *B. dorsalis*, as detected by both the automatic monitoring system and manual monitoring, were found to be congruent (Figure 4). Pearson’s correlation analysis of the daily monitored counts of *B. dorsalis* demonstrated a strong and significant linear correlation between the insect quantities recorded by the two monitoring approaches (Figure 5), suggesting that there is no significant difference between the automated monitoring results by the device and manual monitoring.

### 3.2. Bactrocera dorsalis Occurrence Dynamics in Four Counties

In 2018 and 2019, the adult population of *B. dorsalis* in four counties experienced significant dynamic changes, with activity concentrated in two peak periods each year (Figure 6). In Yuanjiang County, the first peak period was from 12 March to 24 April, during which the daily population reached a peak of 189 insects in 2018. The second peak period was from 26 May to 13 June, with the highest daily incidence reaching 235 insects. In 2019, adult *B. dorsalis* activity commenced in March. From 20 March to 15 April, the population reached its highest recorded level during the two-year study, peaking at 1613 insects daily. The average daily count during this period was 518.96 insects, marking the highest average observed and representing a more than six-fold increase over the previous year. The second peak occurred from 29 May to 9 June, with a daily maximum of 357 insects, significantly lower than the first peak.

In Huaping County, the temperature was lower than that in Yuanjiang (Figure 6). In 2018, the active period of adult *B. dorsalis* typically began in May, and it also had two peak population periods. And the daily maximum numbers of the two peak adult populations were similar, with 211 and 228 individuals, respectively. However, the timing of the first peak population period in Huaping in 2019 overlapped with Yuanjiang and recorded the highest population in both years, with a maximum daily monitored population of 1679 during the first peak period on 11 March. The second peak population period occurred more than two months later. Notably, in Huaping, during the colder month of January, *B. dorsalis* adults were detected on only three days, and no occurrences were noted in February 2019.

The temperatures of Guangnan and Ludian are significantly lower than those of Yuanjiang and Huaping (Figure 6). The number of adults of *B. dorsalis* monitored in Guangnan and Ludian counties was lower than in Yuanjiang and Huaping counties. Guangnan had two distinct peak periods of the *B. dorsalis* population in 2018 and 2019, from May and July to August each year. Adult *B. dorsalis* did not occur in most of December 2018 and 2019 in Guangnan. The monitoring site in the apple orchard in Ludian, which has cooler average year-round temperatures, had the lowest monitored population of the four counties for both years, and the two peaks of *B. dorsalis* occurrence in both years were separated by a short period. In mid-July, the peak population period of *B. dorsalis* in Ludian was the latest among the four monitoring sites. Due to the cold weather, *B. dorsalis* was not monitored from mid-November to mid-May.

### 3.3. Correlation between Meteorological Factors and Population Dynamics of B. dorsalis

Pearson’s correlation analysis was conducted to determine the extent of the association between the adult *B. dorsalis* population and various meteorological factors across different counties (Figure 7). In Yuanjiang County, high temperatures showed a significant positive correlation with the adult *B. dorsalis* population, while humidity demonstrated a significant negative correlation. In Huaping County, all meteorological factors except wind speed showed significant positive correlations with the *B. dorsalis* population. In Guangnan County, low temperature, high temperature, and precipitation showed significant positive correlations. Similarly, in Ludian County, low and high temperatures, precipitation, and humidity were positively correlated with the *B. dorsalis* population, whereas wind speed showed a significant negative correlation. However, the linear correlation between various meteorological factors and the population dynamics of *B. dorsalis* is not strong. In Guangnan and Ludian, the linear relationship between temperature and the abundance of adult *B. dorsalis* is relatively strong (correlation coefficient: Guangnan, low temperature, 0.39, *p <* 0.0001, high temperature, 0.34, *p <* 0.0001; Ludian, low temperature, 0.42, *p <* 0.0001, high temperature, 0.33, *p <* 0.0001).

Correlated weather parameters were grouped into the smallest possible subsets, representing the maximum variation in the original data set, which was conducted by performing principal component analysis (PCA) using 5 variables (Table 1). The analysis reduced these variables to two principal components with eigenvalues greater than 1, explaining 67.8% (Yuanjiang), 73.1% (Huaping), 68.8% (Guangnan), and 75.3% (Ludian) of the total cumulative variance for the entire set of variables (Figure 8). The first component (PC1) exhibits higher factor loadings for low temperature and high temperature in Yuanjiang, Huaping, and Guangnan. In contrast, in Ludian, PC1 demonstrates higher factor loadings for low temperature and wind speed. The second principal component (PC2) displayed varying leading factors across different regions: precipitation and wind speed had higher factor loadings in Yuanjiang, Huaping, and Guangnan, whereas high temperature and humidity were more influential in Ludian County.

Given the seasonal patterns in the occurrence of Bactrocera dorsalis and the identification of temperature as the variable with the highest explained variance through PCA analysis, it was important to determine the temperature thresholds for the initiation of activity and the temperature ranges associated with peak occurrences of adult *B. dorsalis* in each county. In Yuanjiang and Huaping, the population of *B. dorsalis* initially increased and then decreased as the daily average temperature rose. In Yuanjiang, *B. dorsalis* appeared when the temperature reached 18 °C, with numbers exceeding 200 between 24 °C and 32 °C. The population peaked at 25.5 °C and then declined as temperatures continued to rise, remaining low between 33 °C and 36.5 °C (Figure 9). In Huaping, *B. dorsalis* appeared at 15.5 °C, with higher numbers observed between 19 °C and 29 °C. The population decreased after 29 °C and remained low up to 33 °C. In contrast, in Guangnan and Ludian, the number of *B. dorsalis* populations increased with rising temperature. *Bactrocera dorsalis* appeared at 17.5 °C and was most active between 20 °C and 25 °C in these two counties.

## 4. Discussion

### 4.1. Overview of B. dorsalis Population Dynamics in Yunnan

In 2018 and 2019, adult *B. dorsalis* exhibited two distinct active periods in Yunnan. Yuanjiang experienced the earliest peak occurrences, with high-incidence periods beginning in March for both years. In Huaping, a significant rise in *B. dorsalis* was observed in May 2018, but the peak shifted to March in 2019. Additionally, the number of *B. dorsalis* monitored in Yuanjiang and Huaping increased in 2019 compared to 2018. In Guangnan County, the peak occurrence of *B. dorsalis* began in May each year. Conversely, Ludian County saw a later peak, with significant numbers of adult flies first appearing in July annually.

Notably, adult flies were observed throughout the year in Yuanjiang. However, only a few *B. dorsalis* adults were recorded for five days between late December and early February in Huaping. The occurrence of *B. dorsali*s adults in Guangnan was primarily absent during December. Adult *B. dorsalis* in Ludian were recorded from the end of May to the beginning of November in 2018 and 2019. These findings align with historical data on *B. dorsalis* populations [26,30,37].

### 4.2. Impact of Environmental Factors on B. dorsalis Population Dynamics

This study monitored *B. dorsalis* populations in four distinct regions of Yunnan, each with varying climates and host characteristics. Yuanjiang, with its high average annual temperature, exhibited the highest *B. dorsalis* population. When comparing Yuanjiang with Huaping, which had similar orchard plantings, both counties had relatively similar *B. dorsalis* populations in 2018. Yuanjiang’s higher winter temperatures and occasional precipitation likely maintained relatively moist soil conditions, which may have favored the overwintering and eclosion of *B. dorsalis* [38]. This could explain the higher population observed in Yuanjiang in 2019 compared to Huaping. These findings suggest that specific environmental factors, such as temperature and soil moisture levels, significantly influence the population dynamics of *B. dorsalis*.

Nonetheless, in 2019, Yuanjiang experienced more extreme high temperatures in May, with an average monthly high temperature of 38.8 °C, and received minimal rainfall during the spring and summer. These conditions likely contributed to a decrease in the *B. dorsalis* population, particularly during the second peak. Similarly, in Huaping, higher temperatures combined with a sudden increase in precipitation before the second peak in 2019 may have negatively impacted the eclosion of adult *B. dorsalis*, resulting in a reduction and delay in the second occurrence [37]. In contrast, Guangnan and Ludian, characterized by lower winter temperatures, had *B. dorsalis* populations lower than those in Yuanjiang and Huaping. Neither county experienced average daily temperatures exceeding 28 °C during the summer. Guangnan had an average temperature of 23.2 °C from May to September, while Ludian had an average of 19.4 °C. These cooler temperatures contribute to the lower *B. dorsalis* populations observed in these regions than in Yuanjiang and Huaping.

*Bactrocera dorsalis* is attracted to the odor of host fruits, particularly the γ-Octalactone found in mangoes, which induces them to lay eggs within the fruit [39,40]. Additionally, *B. dorsalis* is drawn to the color yellow [41], further propelling them towards ripe fruits. Based on these characteristics, we hypothesize that the fruit tree variety planted at the monitoring sites also significantly influences the population dynamics of *B. dorsalis*. Mango cultivation is a vital industry in Yuanjiang and Huaping, where the mangoes primarily ripen from March to August each year. This period coincides with the heightened occurrence of *B. dorsalis*, posing a substantial threat to the mango industry in these regions. In contrast, Guangnan cultivates citrus fruits that ripen between August and February. The lower temperatures during this period discourage *B. dorsalis* from using unripe citrus fruits as hosts due to their color [4]. In Ludian, apples bloom from May to the end of July and ripen between July and October. Although *B. dorsalis* shows a relatively lower preference for apples compared to mangoes [40], it remains uncertain whether the *B. dorsalis* population in Ludian will evolve to show an increased preference for apples or enhanced cold tolerance. Notably, existing research has demonstrated *B. dorsalis*’s remarkable adaptability to cold stress [42]. Therefore, it is imperative to strengthen prevention and control measures against *B. dorsalis* to avert potential threats to the apple industry in Ludian.

Human factors, such as orchard management, are also crucial influencers in the development of pest populations. However, we have not conducted investigations in this aspect. In the future, we will incorporate more factors into our analysis of the population dynamic changes of *B. dorsalis*, aiming to devise tailored strategies for regional pest management.

### 4.3. Regional Recommendations for B. dorsalis Management

By analyzing the trends in electronic monitoring data for *B. dorsalis* and the climatic variations across different regions, it is possible to devise more precise pest control strategies to improve both the consistency and the amount of fruit produced. During the winter months (December–January), when the population of *B. dorsalis* is low and stable, the control strategy should primarily focus on routine monitoring and low-intensity prevention measures. Specific actions include setting up yellow traps or electronic monitors for surveillance and conducting soil turning or deep mulching in orchards to reduce the survival of overwintering pupae. Before the population increases rapidly, which occurs when the daily mean temperature begins to rise to 17 °C (in February for Yuanjiang and Huaping, April for Guangnan, and June for Ludian), intensified monitoring and control efforts are required. Regular inspections of the pest population in orchards should be conducted. Controlling the adult population is crucial and can be achieved through targeted pesticide applications, timely bagging of fruits, and increasing the number of traps. During the active periods of *B. dorsalis*, effective control can be achieved by destroying the fallen fruits, and employing soil disturbance techniques, such as plowing and ground flooding, after the first and second outbreaks [18,43]. In October, as the temperature decreases, the activity and reproduction of *B. dorsalis* gradually diminish, allowing for a reduction in control efforts. However, it is not advisable to cease prevention measures entirely. This is particularly relevant in Guangnan, where citrus fruits are maturing during this period. In Yuanjiang, where the winter climate is relatively warm, special attention must be paid to the residual pest situation in orchards to prevent a resurgence in the following year. The other three counties have low winter temperatures, which are unsuitable for the overwintering of *B. dorsalis*. Some studies have indicated that *B. dorsalis* populations in Yunnan may not be overwintering but rather migratory [26]. Consequently, these monitoring results provide practical guidance for preemptively setting up adult *B. dorsalis* traps or adjusting control measures, such as fruit bagging, during the winter.

### 4.4. Automated Electronic Monitoring Devices in Pest Monitoring: Current Applications and Future Prospects

The experiment employed an automated system designed to monitor, capture, and count *B. dorsalis* using specific insect parapheromones while transmitting real-time data. This system, combined with meteorological data, enabled the analysis of *B. dorsalis* population dynamics within the monitored area. It also facilitated the prediction of temperature thresholds for *B. dorsalis* activity, providing early warnings of infestations in various locations. Compared to simple lure devices, the primary advantage of this automated system lies in its ability to count insects and transmit data automatically [13]. The monitoring system allowed for customizing monitoring periods based on specific needs and could provide population data as frequently as every minute, a workload impractical for manual counting.

Recent advancements in automated pest monitoring systems have incorporated deep learning techniques for image analysis, significantly enhancing their effectiveness in monitoring target pests [44,45]. This continuous evolution of the monitoring system aims to provide more accurate statistics, the ability to monitor additional climatic data, and the integration and analysis of all available information to enable timely forecasting and prediction. These improvements are crucial for effective *B. dorsalis* control efforts [19,46,47]. Furthermore, intelligent monitoring systems are being developed to enhance user friendliness. These systems can be operated via smartphones and computers, making them accessible to researchers and farmers (Figure 10). This ease of use facilitates widespread adoption and simplifies the process of pest monitoring and control. The electronic monitoring system employed in this study necessitates manual intervention for the monthly addition of attractants. Conversely, the apparatus engineered by Huang et al. [48] for the surveillance of *B. minax* incorporates an automated mechanism for the replacement of yellow sticky paper impregnated with attractants.

From a comprehensive perspective, the intelligent pest monitoring system employed in this study offers accurate, efficient, real-time, practical monitoring data. It lets us promptly grasp the current pest situation and trends, facilitating timely prevention and control measures while reducing labor costs. Moreover, it supports implementing integrated pest management for *B. dorsalis*, reducing pesticide usage. This aligns with intelligent plant protection, emphasizing sustainable and environmentally friendly pest management practices.

## Figures and Tables

**Figure 1 insects-15-00621-f001:**
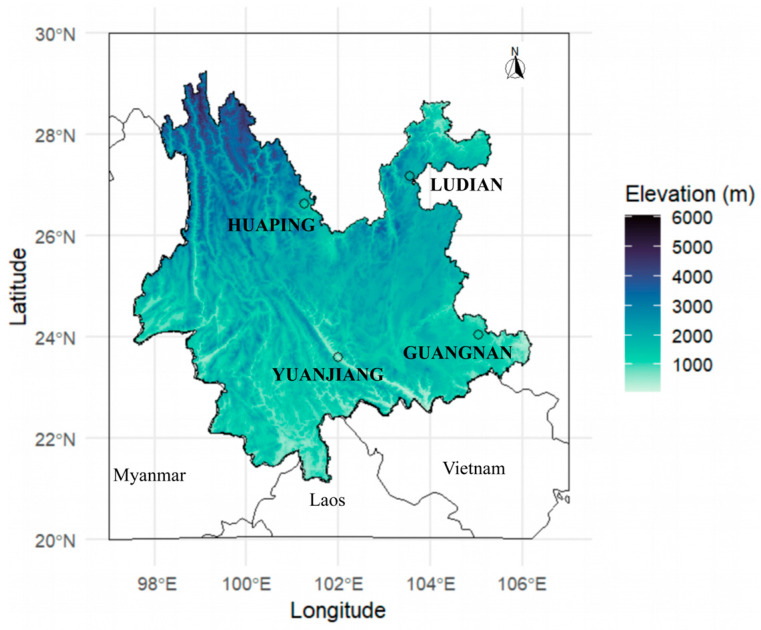
Map of four monitor sites in Yunnan Province. We used the “geodata” package [31,32] to download global elevation data and the R package “ggplot2” [33] to draw the elevation map.

**Figure 2 insects-15-00621-f002:**
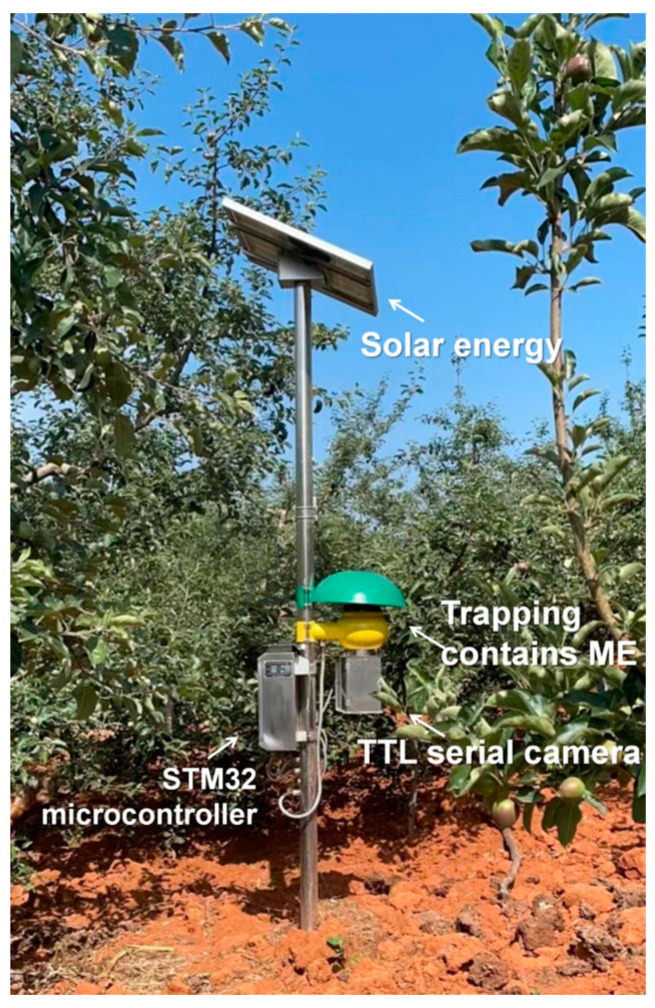
Automatic *B. dorsalis* monitor: TYPE RTJK-1S.

**Figure 3 insects-15-00621-f003:**
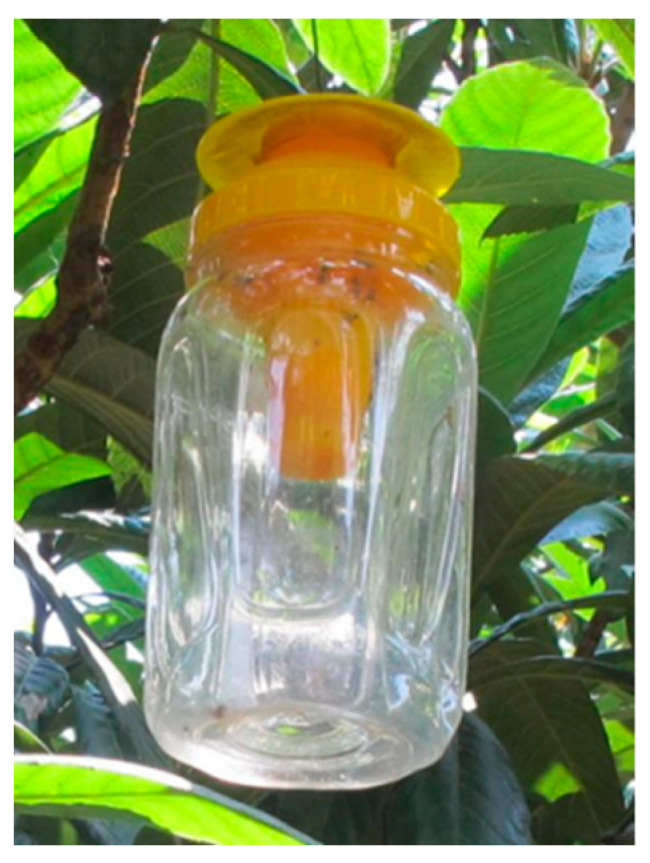
Bottle traps containing the sex parapheromone (ME) for *B. dorsalis.*

**Figure 4 insects-15-00621-f004:**
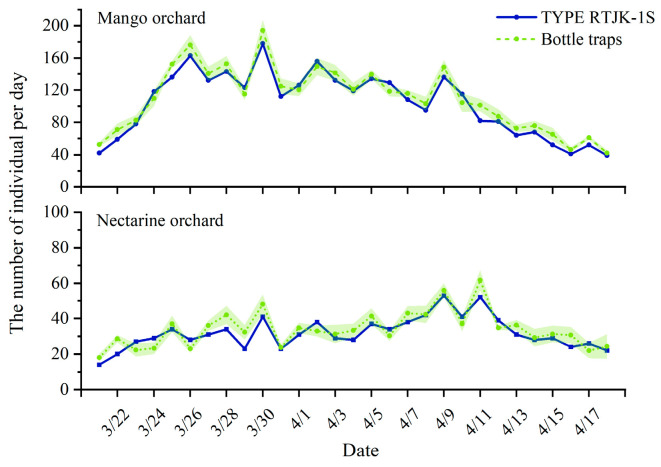
Population dynamics of *B. dorsalis* occurrence from March to April 2017 by using automatic monitoring system and manual monitoring.

**Figure 5 insects-15-00621-f005:**
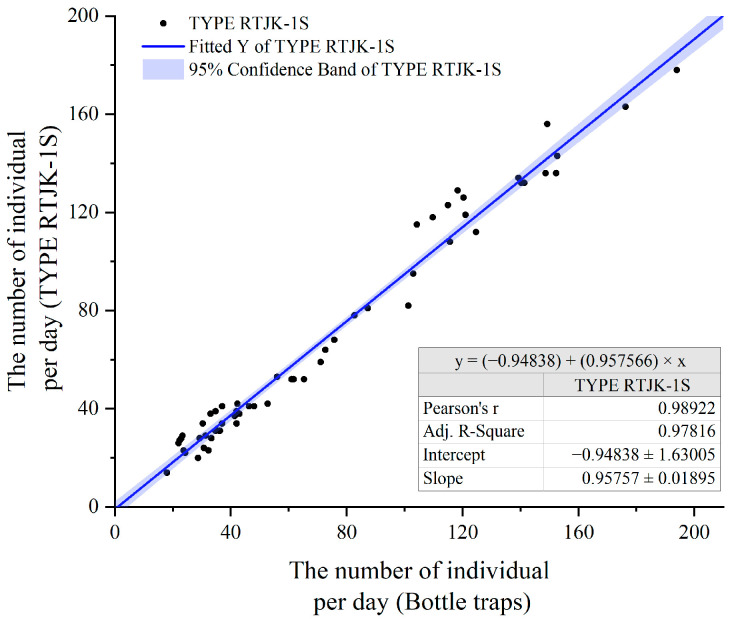
Linear regression analysis of daily *B. dorsalis* captures by automatic system and manual counting.

**Figure 6 insects-15-00621-f006:**
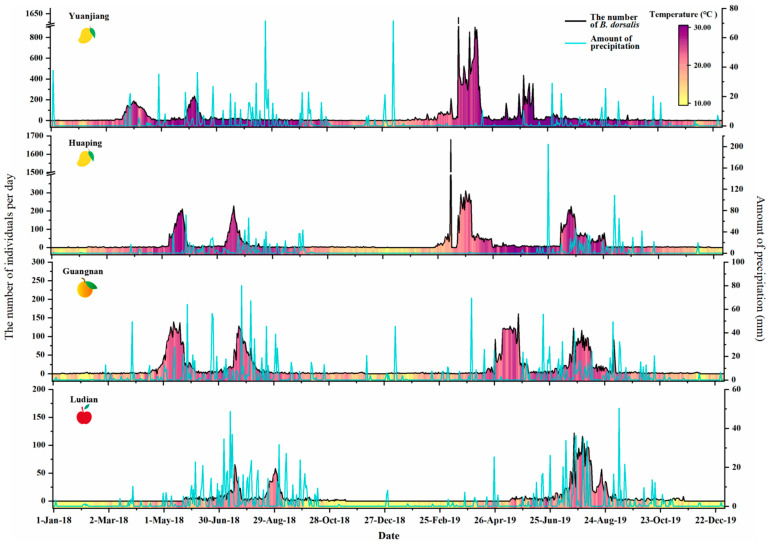
Population dynamics of *B. dorsalis* occurrence in four counties in 2018–2019.

**Figure 7 insects-15-00621-f007:**
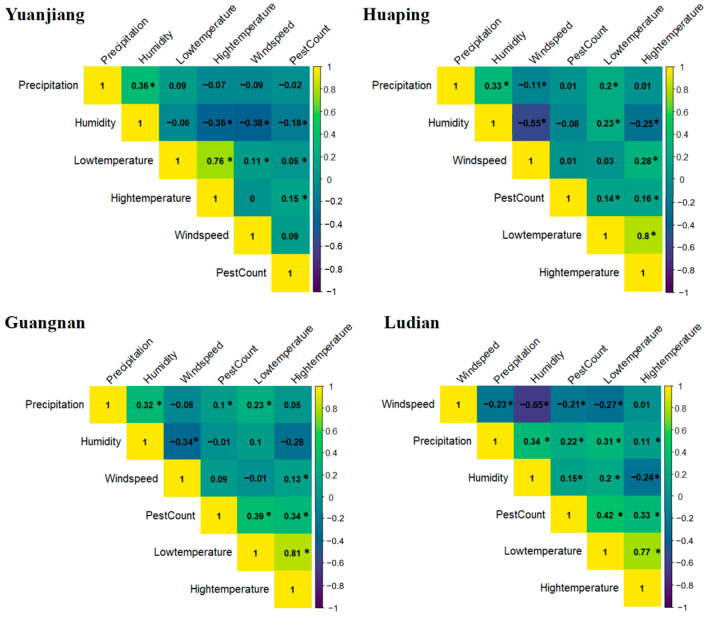
Correlation between the population of *B. dorsalis* and meteorological factors in four counties in 2018–2019. To investigate the relationship between the population of *B. dorsalis* (PestCount) and meteorological factors in four counties during 2018–2019, Pearson’s correlation analysis was employed. A Bonferroni correction was applied to account for multiple comparisons, resulting in a corrected significance level of *p* < 0.0033. Significant correlations are indicated by an asterisk (‘*’).

**Figure 8 insects-15-00621-f008:**
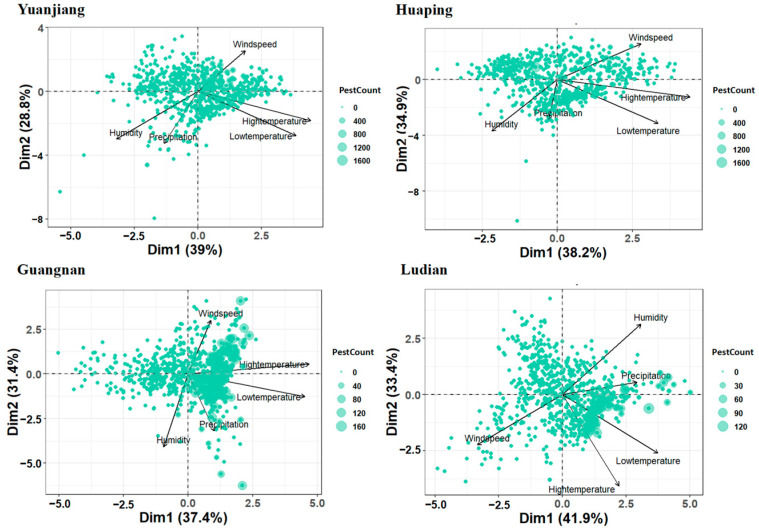
Biplot of principal component analysis (PCA) for meteorological factors and population of *B. dorsalis.*

**Figure 9 insects-15-00621-f009:**
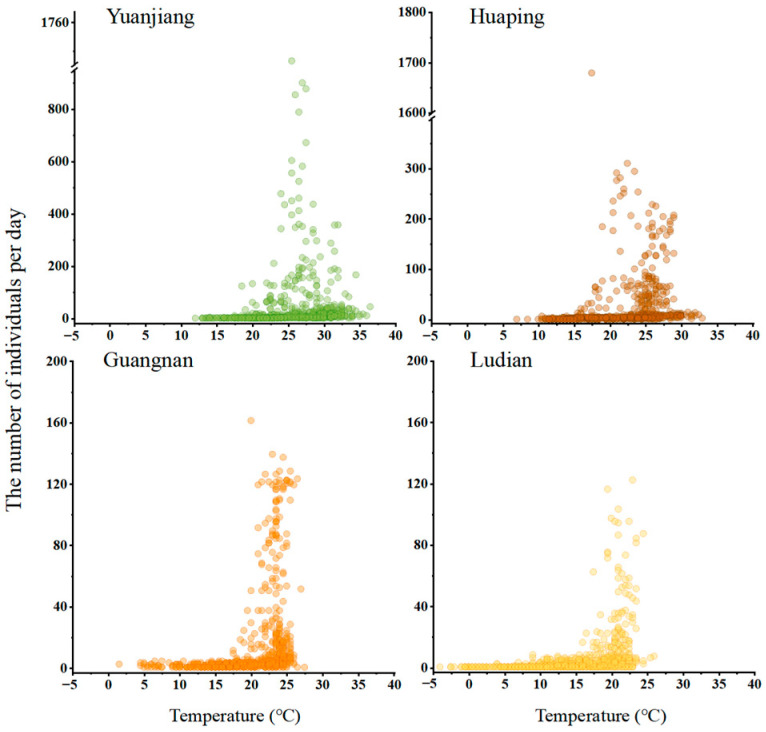
The number of *B. dorsalis* occurrences and average daily temperatures in four counties in 2018–2019.

**Figure 10 insects-15-00621-f010:**
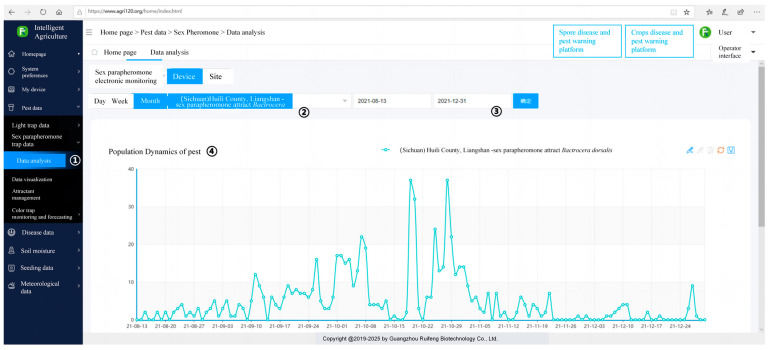
The intelligent agriculture platform, originally in Chinese, provides a computer interface to receive monitoring data. On this webpage, users can locate monitoring devices in the left-hand column. For example, to use data from a sex parapheromone monitoring device, one can select “Sex parapheromone trap data” ①. Next, users can choose the device location ② and the monitoring time period ③. Once these selections are made, the population dynamics of the monitored pests will be displayed on the page ④.

**Table 1 insects-15-00621-t001:** Principal component analysis of weather parameters influencing the number of adult *B. dorsalis.*

Principal Components	Yuanjiang County		Huaping County		Guangnan County		Ludian County	
	PC1	PC2	PC1	PC2	PC1	PC2	PC1	PC2
Eigen values	1.949	1.438	1.910	1.746	1.872	1.570	2.096	1.669
Factor loadings								
Low temperature	0.546	−0.454	0.510	−0.498	0.673	−0.208	0.540	−0.420
High temperature	0.626	−0.298	0.670	−0.197	0.698	0.089	0.319	−0.655
Precipitation	−0.189	−0.535	−0.044	−0.454	0.152	−0.520	0.421	0.088
Wind speed	0.263	0.422	0.424	0.407	0.131	0.488	−0.482	−0.361
Humidity	−0.453	−0.490	−0.332	−0.584	−0.141	−0.663	0.444	0.506

## Data Availability

The data presented in this study are available on request from the corresponding author.

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
