# Peer review of "Population Dynamics of Bactrocera dorsalis (Diptera: Tephritidae) in Four Counties of Yunnan, China, by Electronic Monitoring System"

_insects, 2024, doi:10.3390/insects15080621_

Round 1
Reviewer 1 Report
Comments and Suggestions for Authors
Dear Authors,
It was a pleasure to read this manuscript, with very popular and ongoing subject of the research. Such research is very important at a time when agriculture plays a key role in providing enough food for a growing human population. This is an experimental work providing very valuable results and answers that are more than applicative.
The manuscript is exquisitely written: each segment is concisely formulated, the research subject and results are clearly described and the discussion pointed to the importance of this study. Also, the number of the references is satisfactory, which are mostly recent publications.
I have a few comments and recommendations in order to improve the quality of the manuscript:
1. In the Introduction, you provided only one reference from 2013. regarding the previous use of the afore mentioned methodology for monitoring B. dorsalis. Are there any other examples and evaluations of the effectiveness of that method for counting the mentioned species? Add some additional references.
2. Does this method reliably recognize insects, i.e. is this species easily recognizable and separable from similar species by morphological characters? Do you have any experiences or references with similar comments and observations? If yes, please implement it in the manuscript.
3. Expand the description of the methodology: e.g. how many devices are installed in individual locations, and is this a sufficient number to obtain the overall picture and assess the situation? How close are they installed to the canopy, for pictorial detection to be satisfactory?
4. It would be very useful for non-expert readers to add a paragraph about the biology and the type of nutrition of the species, to make it easier to understand the correlation of population dynamics with the season of the appearance of flowers or ripe fruits.
5. You can add to Discussion e.g. are there plans to continue research and/or introduce new methods in these localities, or expand research to other areas throughout the country? What is the logical continuation of this particular study?
Reviewer 2 Report
Comments and Suggestions for Authors
This article outlines how the authors used an automated
electronic monitoring system to keep track of the population dynamics of the
agricultural pest, Bactrocera dorsalis (Hendel), in four counties
of the Yunnan province, China. Their research confirmed that this system
appears to be accurate in its monitoring capacity, as their findings aligned
with historical data. In addition, though it is unclear where the climatic data
was obtained from, the authors found temperature thresholds at which this fly would
most likely be active, and populations would thrive, enabling the establishment
of warning systems.
While there is value in this research being published there are some serious issues with this article regarding the data analysis, as well as the written structure of this article.
First and foremost, the explicit goals of this research are not clearly stated. The introduction and background appear to be appropriate until the final paragraph of this section. For Lines 78-81, where you state that this monitoring system was assessed, it is unclear how exactly this method was assessed, since as far as I can see, beyond a cursory comparison of your data with historical patterns, I don’t see any methods that assesses whether this method is effective. In addition, lines 82-83 state that “The daily dynamics of B. dorsalis were analyzed with meteorological data from the same year, which provided a scientific foundation for developing a control strategy for B. dorsalis across four counties in Yunnan”. These are not clear goals. Suggestion: You are examining whether there is a correlation between peaks of adult occurrence and meteorological data (temperature and precipitation) and seeing whether these peaks in adult occurrence can be predicted based on these variables (from what I understand) which would allow to establish temperature or precipitation guidelines for control strategies. However, if these are your goals, it is not clear whether you have accomplished the second aspect of your goals.
A second major issue with this article is that the statistical analyses are not appropriate for this data. You are performing multiple tests of correlations and not accounting for it in your alpha value. At a minimum, you should do a correction, such as a Bonferroni correction, to account for the fact that you are comparing 16 tests.
As well, it is not appropriate to do separate correlations for each of the temperature variables, since they are already correlated. One possibility would be to do a principal component analysis for the temperature variables first. However, since you are looking to use these statistics to predict when this fly will be active, this may not be appropriate. Nonetheless, the correlation between these variables needs to be addressed, if not methodologically, then at least in the discussion. It is difficult to make suggestions for improving the statistical methods, when the goals are unclear. I suggest that when you are editing this manuscript, to ensure that you state each goal explicitly, and outline specifically how you are testing/addressing each of these goals, and restructure accordingly.
Specific comments:
The methods, particularly statistical methods need to be more explicit. The description of the sites, and climates at each particular orchard/site which is present in the results, should be in the methods in the first section (Study area).
The map of the study area should have the latitude and longitude axes present. Given better graphical representation, table 1 may not be needed, or at least could contain less information, as the map could show elevation, for instance. Specific comments about table 1 are in the attachment.
In the section 2.2. Monitoring the population dynamics of B. dorsalis, more details are needed. A reference for (so that the reader can refer to it), or an explanation of the Internet of Things technology would be helpful.
Figure 2 is not very helpful without any labels to explain what we are looking at. You could either add labels to this picture or add a diagram of the monitoring system next to it with the different components labelled.
It is not specified in the methods where the precipitation and temperature data were obtained. Was it measured with the monitoring system or was it downloaded? If it was the latter, provide your source. Data downloading could be stated in section 2.3. Data analysis; but if it was obtained through the monitoring system then this could be outlined in section 2.2.
The section 2.3. Data analysis needs more detail. I have provided a suggestion in the attachment as to how to rephrase Lines 115-116 regarding the plotting of the data for Figure 3. However, you should also explain how figures 4 and 5 were created, which is just the number of individuals plotted against a variable, omitting time. The software used for all analyses should be stated.
I would suggest using a different colour palette for all the figures such as viridis if you are using R, or any other palette which accounts for colour blindness.
As mentioned above, the description of the climate should not be in the results section, but rather methods. The results should just describe the population dynamics, and not focus on any explanation.
Lines 192-193 discuss patterns of 18 years and 19 years, and I think this is an error where the patterns should be referring to 2018 and 2019 instead.
The discussion begins with the importance of the monitoring system (lines 206-224), however, since establishing that it is accurate was not a specific goal of this study, I would suggest moving this paragraph to later. I would suggest re-summarizing the results instead and outlining the importance of these results first. There is no review examining whether this is one of the first systems of its kind or whether there are others.
In the discussion, the second paragraph (lines 229- 237) outlines that this system can accurately monitor population dynamics of this fly, but it does not actually demonstrate this in the paper. It does not offer any concrete evidence for this statement. There are no methods, or analyses, or even adequate literature review that examines historical records provided.
Overall, the discussion seems a bit disparate, and subtitles such as: summary, implications, recommendations for pest control might be helpful.
Language:
Careful not to use the word dynamics when you mean population dynamics.
Overall, the writing needs work, particularly the results.
I have attached a pdf of the document with additional specific comments and edits, some of which are outlined above.

Specific comments for edits are outlined above and in the attached document.
Overall, the language needs tightening, especially in the results. It is clear that English is a second language for the authors, so I commend them for their writing, particularly in the introduction (except for goals). Nonetheless, some work is needed to get this manuscript to publication-level.
